# Application of Cmic/Corg in the Soil Fertility Evaluation of Typical Forests in the Yulin Sandy Area

**Yue Wang** [1,2]**, Shan Wang** [1,2]**, Chun-Sheng Zhou** [1,2] **and Wen-Feng Chi** [1,2,*]

1   School of Resources and Environmental Economics, Inner Mongolia University of Finance and Economics, Hohhot 010070, China; wyue@imufe.edu.cn (Y.W.); wangsh@imufe.edu.cn (S.W.); zhouchsh@imufe.edu.cn (C.-S.Z.)
2   Resource Utilization and Environmental Protection Coordinated Development Academician Expert Workstation in the North of China, Inner Mongolia University of Finance and Economics, Inner Mongolia, Hohhot 010070, China
*   Correspondence: cwf@imufe.edu.cn; Tel.: +86-15647151741

**Abstract:** The microbial quotient (Cmic/Corg) has been used extensively to evaluate agriculture soil fertility, but the microbial sensitivity should be considered during the forestry process. Therefore, the objective of this study was to examine a soil fertility evaluation method applied to four vegetation types in the Mu Us Sandland in northwestern China, using the relationship between the Cmic/Corg ratio and soil moisture, and soil temperature under the premise of microbial diversity. The final predictive value was *C. microphylla* (0.2198) > *P. sylvestris* (0.2175) > *P. tabulaeformis* (0.0872) > *S. psammophila* (0.0767). We verified the evaluation results using two traditional methods, the back-propagation (BP) artificial neural network model and principal component analysis, which are widely used to evaluate soil quality based on the soil nutrient concentration. The results were the same as the Cmic/Corg predictions. We conclude that when the soil microbes are used in soil quality evaluations, the changing pattern should be fully considered.

**Keywords:** soil quality; microbial quotient (Cmic/Corg); bacterial diversity

## 1. Introduction

Soil drives the flow of water, energy, and nutrients and supports plant growth. Throughout these processes, soil microorganisms are key drivers of biogeochemical cycling and may be able to rapidly respond to changing environments in ways that alter community structure and functioning [1,2]. Many studies have reported that climate change significantly affects microbial composition and biomass, enzyme activities, and physiological profiles [3,4], although some early studies demonstrated that litter chemistry parameters are the most important drivers of litter decomposition at the ecosystem scale, and the active portion affects the soil microorganisms [5]. These results are not conflicting because, generally, litter decomposition is controlled by the interactions between decomposers and substrate quality, which are both directly affected by environmental variables [6]. Environmental factors, such as soil temperature (ST), are considered to be important regulators of litter decomposition, even at regional scales [7,8].

Despite their important roles, microbial dynamics are only now beginning to be represented in ecosystem models, and we know very little about how microbes respond to changes in microclimates [9,10]. It is difficult to predict how soil microorganisms will respond to the dynamic precipitation patterns caused by environmental changes, but there is no doubt that soil moisture (SM) and temperature play a prominent role in controlling the rates of biogeochemical processes in all terrestrial ecosystems [11]. On the one hand, the SM content can cause soil aggregates to shatter, exposing previously unavailable organic matter for decomposition and releasing nutrients and microbial biomass, possibly due to s microbial hypo-osmotic stress response [12]. On the other hand, ST explains over

90% of the variation in the decomposition rate [7]. Furthermore, warming accelerates soil microbial respiration rates and induces changes in the temperature sensitivity of microbially mediated processes, typically due to increased soil enzyme activities, which drives decomposition [1,13,14]. Although some studies have shown that tree species drive changes in soil microbial biomass composition, as dehydrogenase activity and metabolic quotients have different patterns among different vegetation types [15–17], the essence of this phenomenon is the responses of different vegetation types to SM and temperature changes.

There are two obvious limitations in the above-mentioned studies. One is the limited data on soil microorganisms in desert or sandland environments. Deserts or sandlands constitute the largest biome on Earth, covering over 20% of the global land surface [18]. The characteristics of these environments include low precipitation and high temperatures, which significantly affect soil microorganisms [19]. However, most investigations on the temporal variability of soil microbial communities and soil quality have focused on temperate environments [20,21]. Clearly, ecologically diverse communities should be studied to better understand soil quality via the analysis of soil microorganisms. Another limitation is the lack of awareness of the differences between forest soil and agricultural soil; soil microorganisms are very sensitive to climate change, but this sensitivity differs between the two types of soil because forest soil is more stable and less affected by human activity. The microbial quotient (Cmic/Corg) has been used extensively to evaluate agriculture soil quality but is effective for directly evaluating the quality of forest soil fertility; furthermore, microbial sensitivity should be considered [22,23]. Therefore, we could not use the microbial characters directly because they would change immediately when the soil microclimate changed. The Cmic/Corg ratio could alleviate this contradiction because of the synergistic change. We only need to consider the correlation between the Cmic/Corg ratio and soil microclimate. This method is based on the relationships between Cmic/Corg, soil moisture (SM), soil temperature (ST), and microbial diversity. We verified the evaluation results using two traditional methods, a back-propagation (BP) artificial neural network model and principal component analysis, which are widely used to evaluate soil quality based on the soil's nutrient concentration.

## 2. Materials and Methods

### 2.1. Experimental Site and Soil Sampling

This study was conducted in the southeastern Mu Us Sandland in Yulin, Shaanxi Province, China (109°12′ E, 38°26′ N). This area has a temperate semi-arid continental monsoon climate with an average annual temperature of 7.8–8.6 °C. The annual rainfall is 250–440 mm, with the highest precipitation levels occurring from July to September. Rainfall and warming occur during the same period in this area. Droughts in spring and winter are often combined with intense sandstorms.

Four types of psammophyte plots were established in a desert botanical garden (Pinus sylvestris var. mongholica, Pinus tabulaeformis, Salix cheilophila, and Caragana microphylla). Each plot was $20 \times 20$ m, with three replicates. The mean canopy coverage was 55–83%. The mean stand densities in the arbor were 670 and 1220 N ha$^{-1}$.

A rain-resistant area was set up in each plot. Soil samples were collected simultaneously in these areas at depths of 0–20 cm from 13 to 24 July 2016 after consecutive days of rainfall. There were 144 soil samples collected in total for the assessment of the soil's microbial biomass carbon and nitrogen (MBC and MBN, respectively). At the same time, we sampled the soil, the soil moisture (SM), and soil temperature (ST) data from the 144 samples collected. The final 12 samples on 24 July were used for soil nutrient and microbial diversity analyses. During this period, the atmospheric temperature and moisture were stable. The MBC and MBN concentrations were determined using chloroform fumigation extraction [24]. Soil nutrient characteristics, including available phosphorus (AP), available potassium (AK), available nitrogen (AN), soil organic carbon (SOC), total nitrogen (TN), total potassium (TK), and total phosphorus (TP), were determined according to standard procedures [25]. The microbial community structure was determined by

DNA extraction and 16S rRNA gene amplicon barcode sequencing. DNA was extracted from 12 samples in total: the V3–V4 region of the bacterial 16S rRNA gene was amplified using the universal primers 338F (5'-ACTCCTACGGGAGGCAGCA-3') and 806R (5'-GGACTACHVGGGTWTCTAAT-3'). The reaction mix was as follows: 10× buffer, 5 μL; $Mg^{2+}$ (25 mmol/L), 4 μL; dNTPs (5 mmol/L), 2 μL; primers (10 pmol/L), 3 μL; soil DNA, 10 ng; Taq, 1 μL; and ddH2O, 35 μL. The polymerase chain reaction (PCR) program was as follows: 95 °C for 3 min; 27 cycles of 95 °C for 30 s, 55 °C for 30 s, and 72 °C for 45 s; and a final extension at 72 °C for 10 min.

Amplicons were extracted from 2% agarose gels and purified using the AxyPrep DNA Gel Extraction Kit (Axygen Biosciences, Union City, CA, USA) according to the manufacturer's instructions and quantified using QuantiFluor™-ST (Promega, Madison, WI, USA). Purified amplicons were pooled in equimolar amounts and underwent pair-sequencing (2 × 300) on an Illumina MiSeq platform according to standard protocols.

### 2.2. Statistical Analysis

SPSS software (ver. 18.0; SPSS Inc., Chicago, IL, USA) was used for principal component analysis and for assessing the relationships between Cmic/Corg, SM, and ST. MATLAB 2014b was used for the BP artificial neural network model. Unique sequences were classified into operational taxonomic units (OTUs) with a threshold of 97% identity using UCLUST. Chimeric sequences were identified and removed using Usearch (ver. 8.0.1623). The taxonomy of each 16S rRNA gene sequence was analyzed by UCLUST against the SILVA 119 16S rRNA database using a confidence threshold of 90%.

Different units of the same variables can produce different principal components, and calculations will give more weight to variables with a larger variance while ignoring those with smaller variance. Therefore, we should standardize data before analysis, calculate a covariance matrix of the standardized data and all eigenvalues of the covariance matrix, and determine the number of principal components based on the cumulative contribution rate of eigenvalues and loading values, and the expression of principal components.

### 3. Results

### 3.1. Bacterial Diversity and Species Composition Analyses

Sequence reads were binned into OTUs based on 97% sequence identity. From a total of 60,091 OTUs, after the removal of singletons, doubletons, and chimeras, the remaining sequences were binned into 15,509 OTUs from 12 samples. Although the sample number was small, the species accumulation and Shannon diversity indices reached saturation, indicating that the majority of the bacterial species were recovered (Figure 1a). This showed that the 12 soil samples were representative of the microbial community structure conditions of the four vegetation types. The Good's coverage was >90% for all samples, indicating that a great majority of species were recovered (Table 1). The alpha-diversity indices of the samples all showed that the *P. sylvestris* stand soil had the most bacterial species but without a pattern of the arbors and shrubs (Table 1). However, there was no significant difference between vegetation types.

**Table 1.** Samples' alpha-diversity.

|  | Chao1 | Good's Coverage | Shannon | OTUs |
|---|---|---|---|---|
| *P. sylvestris* | 7503.16 ± 414.189 | 0.918 ± 0.007 | 9.48 ± 0.37 | 5036 ± 742 |
| *C. microphylla* | 6130.645 ± 1305.308 | 0.933 ± 0.016 | 8.68 ± 0.73 | 4004 ± 627 |
| *S. cheilophila* | 6356.704 ± 1136.533 | 0.931 ± 0.013 | 9.12 ± 0.61 | 4635 ± 526 |
| *P. tabulaeformis* | 6231.064 ± 898.552 | 0.933 ± 0.010 | 9.25 ± 0.42 | 4372 ± 662 |

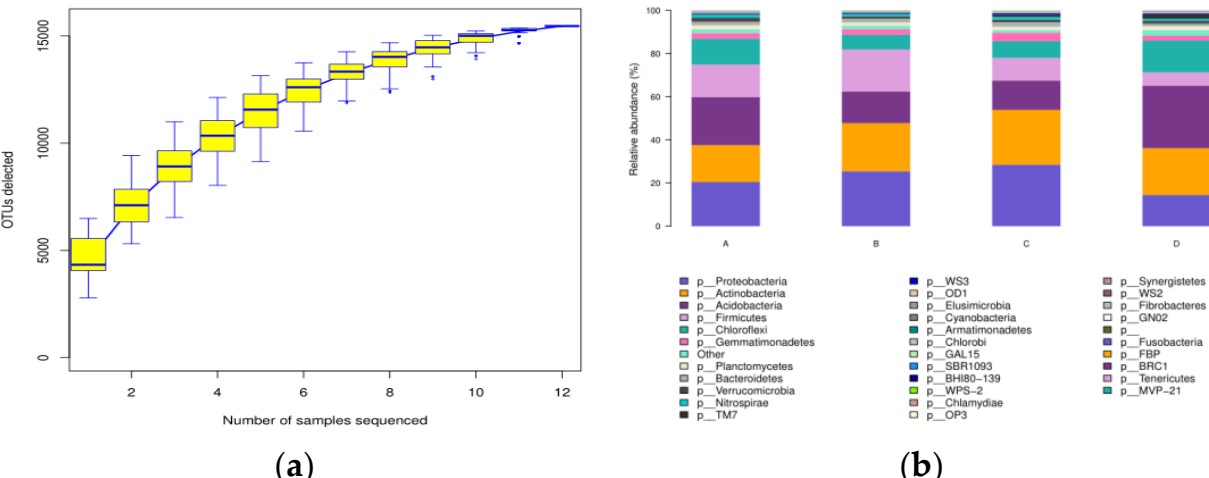

(**a**)                                          (**b**)

**Figure 1.** Species accumulation curve and Shannon diversity indices. (**a**): Species accumulation curve, (**b**): phylum-level taxonomic distribution of bacterial operational taxonomic units (OTUs) (97% cutoff). A: *P. sylvestris*; B: *C. microphylla*; C: *S. cheilophila*; D: *P. tabulaeformis*.

The top five bacterial phyla were the same for all four vegetation types, Proteobacteria, Actinobacteria, Acidobacteria, Firmicutes, and Chloroflexi (Figure 1b), but the predominant phyla were different. Acidobacteria were dominant in *P. sylvestris* (22.32%) and *P. tabulaeformis* (29.02%) samples, whereas Proteobacteria were dominant in C. microphylla (27.64%) and S. cheilophila (28.51%) samples.

### 3.2. Soil Quality Evaluation Based on Changes in the Cmic/Corg Ratio

There were no significant differences in the microbial diversity index and dominant species abundance between the four stands, which could show that the effect of the microbial community structure on the microbial biomass carbon (MBC) was not different among the stands. Therefore, in the evaluation of soil fertility quality, there is no need to put the eigenvalue of the soil microbial community structure into the models.

After consecutive days of rainfall, the sampling and measurement of MBC, MBN, ST, and SM were conducted for 12 days without water input. Although the SM had a decreasing trend, the ST and SM showed no obvious changes in pattern, especially ST. MBC and MBN responded differently after cessation of water input; the MBC decreased immediately and rapidly until the tenth day, after which it stabilized while maintaining a decreasing trend. The response of MBN was slower; the pattern of MBN changed just like a trapezoid. This difference indicated that MBC is more sensitive to SM changes and that carbon is more dynamic than nitrogen in soil. The most stable factor after the cessation of water input was Cmic/Corg, which stabilized after the fourth day without rainfall, and the tendency was approximately the same for all four vegetation types (Figure 2).

We used six curves (cubic, linear, s-shaped, logical, quadratic, and logarithmic) to model the relationships among Cmic/Corg, SM, and ST. Variables were retained in the model when the F-test statistic was significant; otherwise, variables were removed. All of the included factors were tested repeatedly, with t-tests being used to ascertain whether the factors and constants in the model equations were significant (Table 2). The result showed that arbors and shrubs had different responses to the soil environmental factors.

**Table 2.** Regression curve coefficients.

| | *P. sylvestris* | *C. microphylla* | *S. cheilophila* | *P. tabulaeformis* |
|---|---|---|---|---|
| Factor (X) | SM | ST | ST | SM |
| Pearson Correlation | 0.720 | 0.353 | 0.420 | 0.374 |
| Sig. | 0.000 | 0.091 | 0.041 | 0.074 |
| Curve | Cubic | Linear | S-shaped | Logical |
| AR$^2$ | 0.632 | 0.084 | 0.177 | 0.119 |
| Equation (Y = Cmic/Corg) | $Y = 0.176 - 0.001X^2 + (5.716 \times 10^{-5})X^3$ | $Y = 0.11X + 0.018$ | $Ln(Y) = -0.807 - 38.564/X$ | $Ln(1/Y) = 44.294 + 0.935\,X$ |

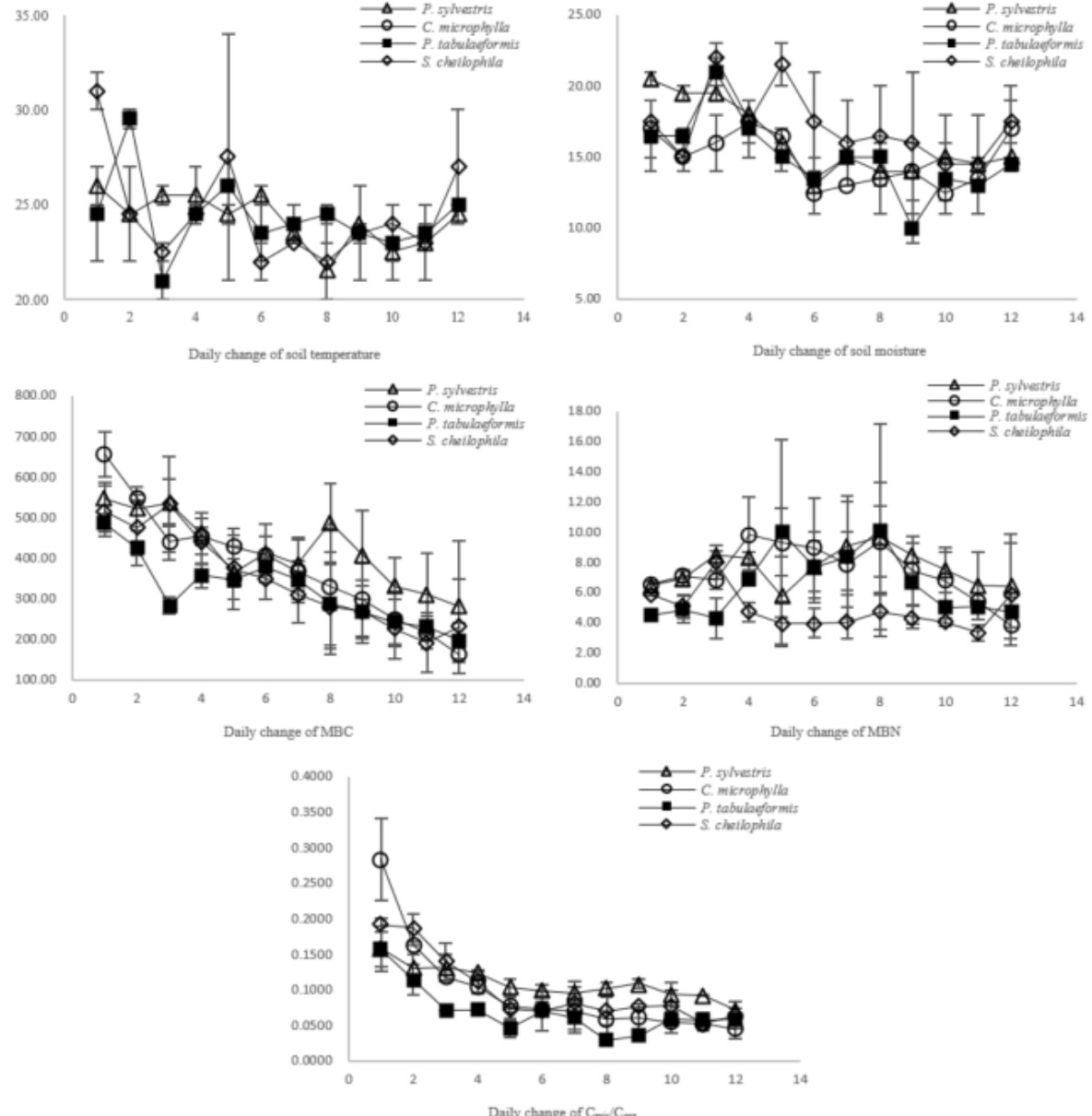

**Figure 2.** Changes in soil temperature (°C) soil moisture (%), microbial biomass carbon (mg/kg), microbial biomass nitrogen (mg/kg), and the microbial quotient (Cmic/Corg).

It is difficult to simultaneously monitor the SM and ST of many plots in the field, so we used the atmospheric temperature (AT) and time after cessation of the water input (interval days, ID). The ST could be calculated based on the average ST/AT ratio during the sampling days of different vegetation types (*P. sylvestris* = 1.01, *C. microphylla* = 1.05, *S.*

*cheilophila* = 1.02, *P. tabulaeformis* = 1.01), and the SM could be calculated based on the linear change of SM with the interval days (Figure 3) and data from weather station monitoring.

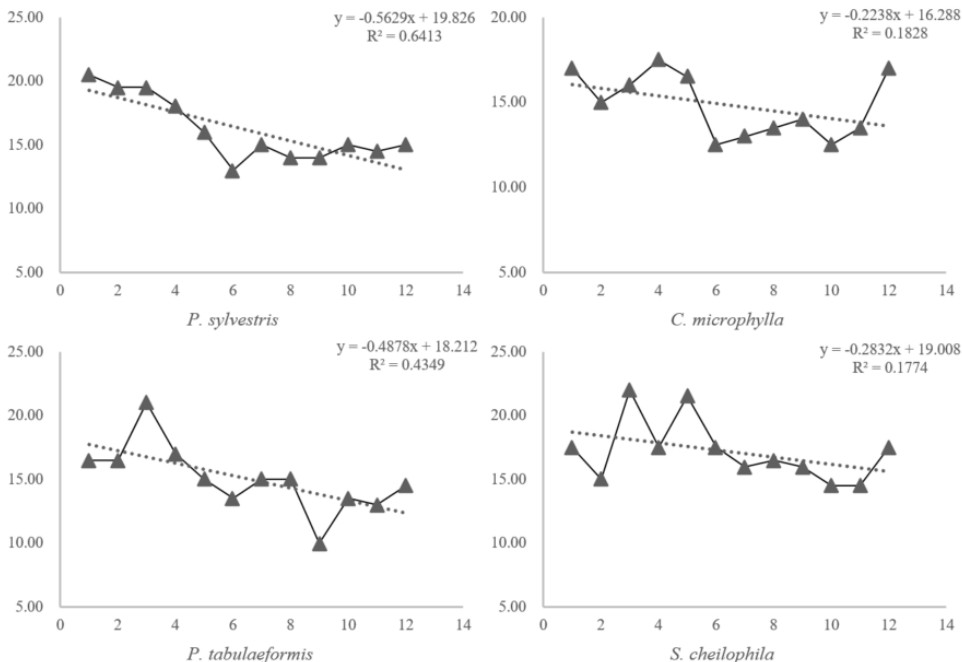

**Figure 3.** Soil moisture (%) changes with the interval days.

The final equations for Cmic/Corg, AT, and ID are shown in Table 3. ID∈[1,7], and AT∈[15,30], with one interval value. All Cmic/Corg predictions were averaged and the soil quality order was *C. microphylla* (0.2198) > *P. sylvestris* (0.2175) > *P. tabulaeformis* (0.0872) > *S. cheilophila* (0.0767).

**Table 3.** Final prediction models of Cmic/Corg.

| | *P. sylvestris* | *C. microphylla* | *S. cheilophila* | *P. tabulaeformis* |
|---|---|---|---|---|
| Original factor | SM | ST | ST | SM |
| Final factor | ID | AT | AT | ID |
| Independent replaced | SM = −0.5629ID + 19.826 | ST = 1.05AT | ST = 1.02 AT | SM = 0.4878ID + 18.212 |
| Final equation | $Y = 0.5787 - 0.0103X^2 - 0.0223X$ | $Y = 0.1155X + 0.0188$ | $Y = 1/[0.4462EXP(37.8078/X)]$ | $Y = 1/[0.2336EXP(0.4561X)]$ |

### 3.3. Verifying the Evaluation Results Using a BP Artificial Neural Network Model and Principal Component Analysis

The soil fertility characteristics are shown in Table 4. The cumulative contribution rate of the first four main components reached 90.30%, and the factors were seven soil nutrient indicators, which reflect the basic fertility quality of the plot (Table 5). The soil fertility quality composite score for different vegetation types can be obtained based on Equations (1)–(4).

$$F1 = 0.255X1 + 0.279X2 - 0.286X3 + 0.222X4 + 0.090X5 - 0.200X6 + 0.017X7 \quad (1)$$

$$F2 = -0.205X1 - 0.083X2 - 0.072X3 + 0.256X4 - 0.509X5 + 0.227X6 + 0.542X7 \quad (2)$$

$$F3 = -0.188X1 + 0.314X2 - 0.118X3 - 0.255X4) + 0.562 X5) + 0.287X6 - 0.582X7 \quad (3)$$

$$F4 = 0.465X1 + 0.063X2 - 0.413X3 - 0.510X4) - 0.193 X5) + 0.645X6 + 0.323 X7 \quad (4)$$

**Table 4.** Soil fertility characteristics of different vegetation types.

|  | TN (g/kg) | TP (g/kg) | TK (g/kg) | AN (mg/kg) | AP (mg/kg) | AK (mg/kg) | SOC (g/kg) |
|---|---|---|---|---|---|---|---|
| *P. sylvestris* | 0.19 ± 0.01 | 0.25 ± 0.00 | 22.48 ± 0.30 | 17.50 ± 1.08 | 1.06 ± 0.05 | 89.50 ± 14.78 | 0.07 ± 0.01 |
| *C. microphylla* | 0.21 ± 0.01 | 0.28 ± 0.17 | 22.03 ± 0.09 | 22.16 ± 0.12 | 1.45 ± 0.20 | 82.17 ± 6.26 | 0.09 ± 0.02 |
| *S. cheilophila* | 0.17 ± 0.26 | 0.23 ± 0.01 | 23.00 ± 0.39 | 17.44 ± 0.70 | 0.68 ± 0.09 | 139.58 ± 12.58 | 0.08 ± 0.00 |
| *P. tabulaeformis* | 0.14 ± 0.01 | 0.23 ± 0.00 | 23.00 ± 0.12 | 18.95 ± 0.65 | 1.60 ± 0.60 | 92.25 ± 26.76 | 0.08 ± 0.00 |

TN, total nitrogen; TP, total phosphorus; TK, total potassium; AN, available nitrogen; AP, available phosphorus; AK, available potassium; SOC, soil organic carbon.

**Table 5.** Total variance explained.

| Component | Initial Eigenvalues | | | Extraction Sum of Squared Loadings | | |
|---|---|---|---|---|---|---|
|  | Total | % of Variance | Cumulative % | Total | % of Variance | Cumulative % |
| 1 | 3.100 | 44.284 | 44.284 | 3.100 | 44.284 | 44.284 |
| 2 | 1.341 | 19.154 | 63.438 | 1.341 | 19.154 | 63.438 |
| 3 | 1.053 | 15.040 | 78.478 | 1.053 | 15.040 | 78.478 |
| 4 | 0.827 | 11.819 | 90.298 | 0.827 | 11.819 | 90.298 |
| 5 | 0.443 | 6.323 | 96.621 |  |  |  |
| 6 | 0.148 | 2.120 | 98.740 |  |  |  |
| 7 | 0.088 | 1.260 | 100.000 |  |  |  |

Components explained are the TN (total nitrogen), TP (total phosphorus), TK (total potassium), AN (available nitrogen), AP (available phosphorus), AK (available potassium), and SOC (soil organic carbon), which can be combined into four principal components (90.298%).

The final scores and the order of the soil quality were *C. microphylla* (0.639 ± 0.202) > *P. sylvestris* (−0.015 ± 0.240) > *P. tabulaeformis* (−0.377 ± 0.322) > *S. cheilophila* (−0.247 ± 0.419). These results were the same as the Cmic/Corg predictions. Although the order was the same, the standard error was too high for the results to be credible. Therefore, we used a BP artificial neural network model to verify the result again. Because there was a small amount of raw data, the first step in training the BP artificial neural network was to generate data randomly based on the soil nutrient classifications in the second national soil survey of China (Table 6). The hidden layer uses a Sigmoid function, and the output layer uses the purelin function. The structure of the BP artificial neural network and the training results are shown in Figure 4.

**Table 6.** Soil nutrient classification.

| Classification | TN (g/kg) | TP (g/kg) | TK (g/kg) | AN (mg/kg) | AP (mg/kg) | AK (mg/kg) | SOC (g/kg) |
|---|---|---|---|---|---|---|---|
| 1 | >2 | >1 | >25 | >150 | >40 | >200 | >40 |
| 2 | 1.5–2.0 | 0.8–1 | 20–25 | 120–150 | 20–40 | 150–200 | 30–40 |
| 3 | 1.0–1.5 | 0.6–0.8 | 15–20 | 90–120 | 10–20 | 100–150 | 20–30 |
| 4 | 0.7–1.0 | 0.4–0.6 | 10–15 | 60–90 | 5–10 | 50–100 | 10–20 |
| 5 | 0.5–0.7 | 0.2–0.4 | 5–10 | 30–60 | 3–5 | 30–50 | 6–10 |
| 6 | <0.5 | <0.2 | <5 | <30 | <3 | <30 | <6 |

TN, total nitrogen; TP, total phosphorus; TK, total potassium; AN, available nitrogen; AP, available phosphorus; AK, available potassium; SOC, soil organic carbon. The c shows the soil was fertility grade.

The trained BP neural network model was used to simulate the test samples; the eight hidden layers and the final soil nutrient classification results can be calculated based on Equations (6)–(14).

$$h1 = 0.8224 + 0.5333TN + 0.6194TP - 0.1470TK - 0.3230AN - 0.0991AP + 0.0459AK + 0.7329SOC \tag{6}$$

$$h2 = -0.4376 + 0.6862TN + 0.5952TP + 0.8443TK - 0.2781\,AN - 0.5083\,AP - 0.0562\,AK + 0.8901SOC \tag{7}$$

$$h3 = 0.1884 - 0.6919TN - 0.8079\,TP + 0.5676TK + 0.3595AN - 0.7153AP + 0.6485AK - 0.4911SOC \tag{8}$$

$$h4 = -0.4459 + 0.9467TN + 1.1603TP + 0.8942TK + 0.3826AN + 0.3488AP + 0.4826AK + 0.0170SOC \tag{9}$$

$$h5 = 0.6558 + 0.4864TN + 0.9807TP + 0.2563TK + 0.8164AN + 0.1803AP - 0.5727AK - 0.1577SOC \tag{10}$$

$$h6 = -0.4126 - 0.7090TN + 0.1058TP - 0.9437TK - 0.2596AN - 0.5530AP - 0.0705AK - 0.9140SOC \tag{11}$$

$$h7 = -0.0883 - 0.5577TN + 0.3895TP + 0.7176TK - 0.1564AN + 1.1792AP - 0.0092AK - 0.4447SOC \tag{12}$$

$$h8 = 0.4541 + 0.1437TN - 0.5153TP + 0.8654TK - 0.5007AN - 1.1573AP + 0.1708AK - 0.1866SOC \tag{13}$$

$$H = -0.2468 - 0.3975h1 - 0.2959h2 - 0.7147h3 - 0.0091h4 - 0.8401h5 - 0.0168h6 - 0.0756h7 + 0.3131h8 \tag{14}$$

The final soil nutrient classification results were *S. cheilophila* (5.0160 ± 0.0168) > *P. tabulaeformis* (5.0096 ± 0.0384) > *P. sylvestris* (4.9774 ± 0.0288) > *C. microphylla* (4.8982 ± 0.0255). The soil quality increases with decreasing nutrient classification, so the order of the soil quality of the four vegetation types was the opposite. These results were also identical to the Cmic/Corg predictions.

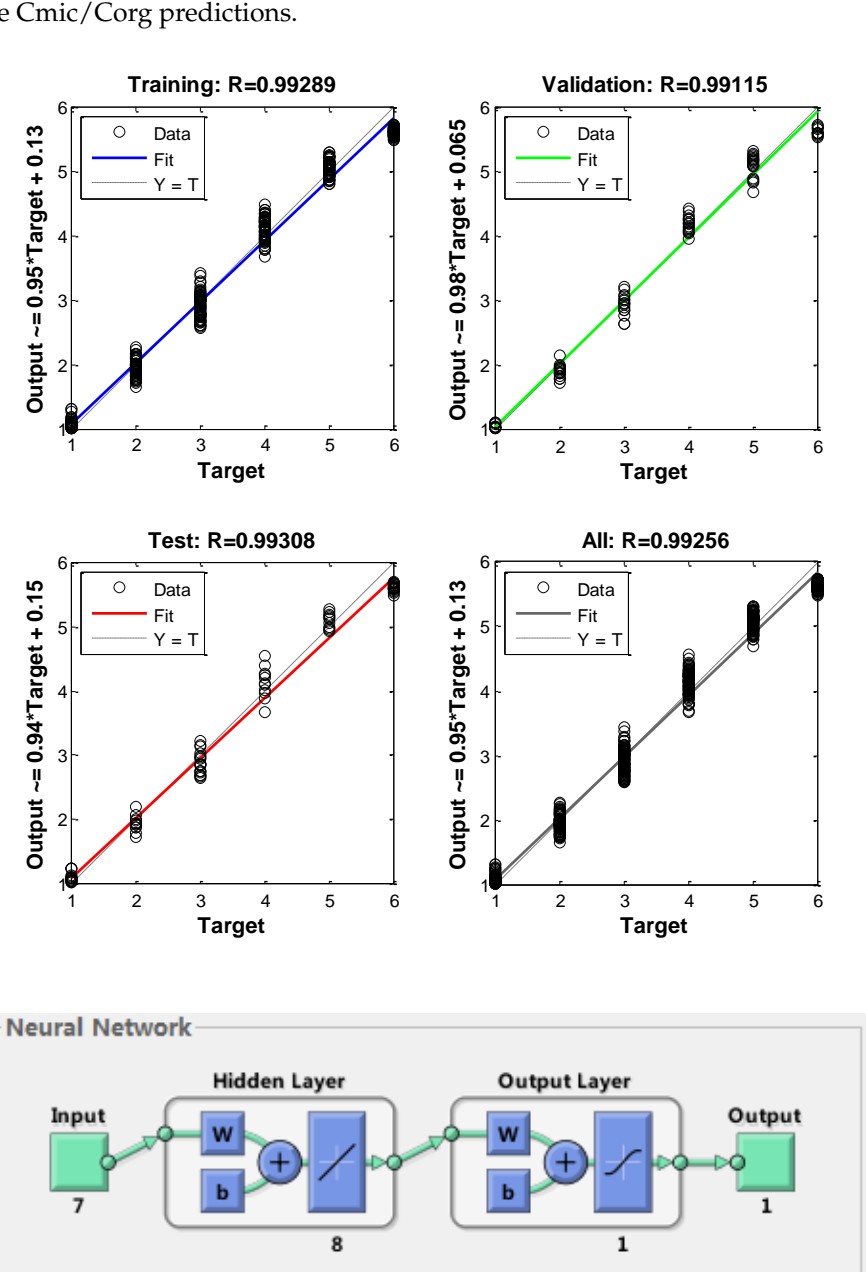

**Figure 4.** Structure of the back-propagation (BP) artificial neural network and training results.

## 4. Discussion

In this study, the species composition of the soil microbial analysis results was similar to those of previous studies [26–28], indicating that common members of these phyla are probably well adapted to survive and possibly thrive in desert soils. However, changes in the abundances of bacterial phyla could represent the response of the microbial community after the soil environmental factors changed; the impact on soil fertility quality is questionable, as the quantity and validity of a bacterial species are not necessarily related. Although the question of whether changes in bacterial diversity and species composition could indicate a change in soil quality requires more discussion, it is certain that they can indicate environmental changes. Proteobacteria and Actinobacteria were more abundant in shrub soil than in arbor soil because most Proteobacteria, as well as the parasitic Actinobacteria, are anaerobic; the litter layer of arbor soil was much thicker than that of shrub soil, resulting in inferior ventilation and generating a more anaerobic environment. Acidobacteria were the opposite; the reason is controversial, but one speculation is that it may be related to root distribution [29].

ST and SM are two factors that impact the observed responses of the Cmic/Corg ratio because warming and rewetting may promote nutrient cycling directly by increasing soil and litter decomposition rates and net N mineralization [30,31]. Several studies proposed that this could also be explained by the labile C content, which is influenced by increased below-ground biomass and higher substrate inputs [32–34]. Based on the results from the present study, soils with different vegetation types have different responses to ST and SM changes under the same rainfall and atmospheric temperature conditions. This is indicative of differences in the buffer capabilities of different plant community compositions under environmental changes. Soil fertility quality indicates the long-term ability to support plant growth; the sensitivity of the MBC reflects changes in the environment rather than changes in soil quality. The Cmic/Corg ratio was stabilized in a short time and fully reflected the proportion of active organic carbon in the soil. The most important point is that Cmic/Corg could explain the difference in the response of soil to environmental changes due to both microbiology and fertility, but it was also too sensitive to evaluate soil quality, so the prediction models in this study provide a feasible way to make a sensitive factor more scientific to show a long-term characteristic. The results of the BP artificial neural network model and principal component analysis verified the rationale of soil fertility quality evaluation based on pattern changes in the Cmic/Corg ratio.

## 5. Conclusions

Using a very sensitive factor as an indicator of the ability of soil to support plant growth after a long-term change is challenging, and it is unreasonable to judge soil quality solely based on the responses of microorganisms to environmental factors; a more stable edaphic factor is required, i.e., one that can cover soil microbial characteristics and consider sensitivity and stability at the same time. Our method has limitations, for example, the model could be more complex instead of linear, and the relationship between the Shannon diversity index and Cmic/Corg ratio should be further explored. However, the present results provide a platform for further studies. When the soil microbes are used in soil quality evaluations, the change pattern should be fully considered.

**Author Contributions:** Conceptualization, Y.W.; methodology, Y.W.; software, Y.W. and S.W.; formal analysis, C.-S.Z.; investigation, W.-F.C.; funding acquisition, W.-F.C. All authors have read and agreed to the published version of the manuscript.

**Funding:** This research was funded by the Natural Science Foundation of Inner Mongolia (NO. 2020BS03001) and the National Natural Science Foundation of China (No. 42061069).

**Informed Consent Statement:** Informed consent was obtained from all subjects involved in the study.

**Data Availability Statement:** All data in this study were obtained experimentally. If the journal requires it, the author can be contacted to provide the original data.

**Acknowledgments:** All authors would like to express our sincere thanks to Beijing Forestry University for its support of this research.

**Conflicts of Interest:** The authors declare no conflict of interest.

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
