# Peer review of "Application of Cmic/Corg in the Soil Fertility Evaluation of Typical Forests in the Yulin Sandy Area"

_land, doi:10.3390/land11040559_

Round 1

Reviewer 1 Report

The publication should be modified.

  1. The entire manuscript is generally poorly written. There are a lot of editorial fixes. The manuscript should be written using the Microsoft Word template.
  2. There is now incorrectly cited literature in the text.
  3. In chapter 2. Materials and methods, provide more information (e.g. standards, used apparatus) on the determination of available phosphorus (AP), available potassium (AK), available nitrogen (AN), soil organic carbon (SOC), total nitrogen (TN), total potassium (TK), and total phosphorus (TP).
  4. Change Figure 2 to another type as it is now unreadable.
  5. The changes are not visible now.
  6. List the homogeneous groups in table 4.
  7. Chapter “4. Discussion” must be extensive.
  8. Now it is very weak.
  9. The Reference chapter is very badly written.
  10. You absolutely need to correct it. Please read the guidelines for the authors of this journal carefully.
  11. Please, be sure that all the references are included in the reference list and vice versa with matching spellings and dates.

Author Response

Dear editor:

Thank for the experts’ revision opinions, which have been revised one by one according to the expert opinions.

1, The manuscript has been written again using Microsoft Word template.

2, The determination of soil nutrition were conventional and known in the academia, it would take up a lot of space, so we decided to make a simple description

3, the Fig to was re-upload as jpg. Type

4, There is no significant difference between vegetation type of soil nutrition, so the homogeneous don’t have sense in the manuscript

5, Reference format and citation location has been confirmed.

6, Other modifications have been marked in the revision manuscript

Reviewer 2 Report

Dear colleagues,

the manuscript contain some interesting results, but the organization of the text will be deeply revisioned. Please see the attached file for the comments.

The title and the introduction is not well organized. The experimental design seems correct, but in my opinion the experimental lenght is too short for a real description of the phenomena studied, and the data elaboration is not well reported. The results chapter is really bad written and a complete revision is required. The quality of figures and tables in insufficient. The discussion is superficial and not usefull for readers.

Best regards.

Author Response

Dear editor:

Thank for the experts’ revision opinions, which have been revised one by one according to the expert opinions.

1 The title has been revised

2 The abstract has rewritten, we added some important information which the reviewer pointed.

3 In the part 1, paragraph 3, we explained the importance of Cmic/Corg ratio as the key factor on the soil fertility, and added some references

4 We added the data sources of atmospheric temperature and moisture

5 We adjusted the position of some paragraphs in the manuscript followed suggestion of reviewer

6 The t-test has been added in table 2

7 The factors of PCA has been pointed in 3.3, paragraph 1

8 Other modifications have been marked in the revision manuscript

Reviewer 3 Report

Comments and suggestions for Authors

 „Application of Cmic/Corg in soil fertility evaluation of typical forests in Yulin Sandy Area”

Subject is very interesting and fall within the scope of the journal. The experimental dataset undoubtedly are useful and constitutes scientific values. The presented manuscript deals with the current local problem.

The aim of the research was to examine a soil quality evaluation method applied to four vegetation types in Yulin, Mu Us Sandland, NW China, using the Cmic/Corg ratio.

General remarks

In order to increase the usefulness of the article, Authors must refer to the following points.

  • The abstract and the introduction are factually correct and do not raise my objections.
  • The section "Materials and methods" should be completed. The chemical properties of the soil should be reported before starting the tests. The distribution of precipitation and temperature during the experiment should be presented. In what years was the experiment conducted? A well-seen photo of a field experiment being conducted. The principle of marking MBC and MBN should be described. Extraction reagents for the determination of AP, AK, AN, SOC, TN, TK and TP should be completed.
  • The obtained results were described correctly.
  • The discussion and conclusion were presented correctly and reliably.
  • References made very carelessly. Please save correctly in accordance with editorial requirements.

Specific comments

Page 1 and 2 – Introduction: In place of the cited Authors, the numbering in the list of references should be used.

Pages 5 – 6 Figures 2 and 3 -You need to complete the units on the 0Y axis.

Page 7 – He et al.,  2009  (not listed in References).

Page 7 – Table 5 Below table 5 write explanations: Component 1-7.

Page 8 – Table 6 Below table 6 write explanations: Classification 1-6.

The entire manuscript must be corrected according to the editorial requirements of the publisher.

Author Response

Dear editor:

Thank for the experts’ revision opinions, which have been revised one by one according to the expert opinions.

1 In place of the cited Authors, the numbering in the list of references has been used.

2 In order to make the figures more clear, the units on the 0Y axis of Figures 2 and 3 has been added in the title of figures.

3 The explanations of table5,6 has been added.

4 Other modifications have been marked in the revision manuscript

Round 2

Reviewer 1 Report

It's much better than the earlier version. I accept it for publication.

Author Response

dear reviewer:

I especially appreciate your being able to agree to accept my manuscript

                                                                                                        best wishes

                                                                                                         Wang Yue
